# Integrative Transcriptomic and Phytohormonal Analyses Provide Insights into the Cold Injury Recovery Mechanisms of Tea Leaves

**DOI:** 10.3390/plants11202751

**Published:** 2022-10-18

**Authors:** Zhi-Qi Ni, Jing Jin, Ying Ye, Wen-Wen Luo, Ya-Nan Zheng, Zheng-Kun Tong, Yi-Qing Lv, Jian-Hui Ye, Liang-Yu Wu

**Affiliations:** 1Tea Research Institute, Zijingang Campus, Zhejiang University, Hangzhou 310013, China; 2Zhejiang Agricultural Technical Extension Center, 29 Fengqi East Road, Hangzhou 310020, China; 3Jinhua Department of Economic Specialty Technology Promotion, 828 Shuanglong South Road, Jinhua 321000, China; 4Lanxi Chishan Lake Green Farm Co., Ltd., Lanxi 321100, China; 5College of Horticulture, Fujian Agriculture and Forestry University, 15 Shangxiadian Road, Fuzhou 350002, China

**Keywords:** *Camellia sinensis*, cold injury recovery, windbreak, phytohormone, transcriptome, transcription factor

## Abstract

Tea plant is susceptible to low temperature, while the cold injury recovery mechanisms of tea leaves are still unclear. Windbreak has an effective and gradient range of protecting tea plants. Tea plants with increasing cold damage degree have varying recovery status accordingly, which are the ideal objects for investigating the cold injury recovery mechanisms of tea leaves. Here, we investigated the transcriptome and phytohormone profiles of tea leaves with different cold injury degrees in recovery (adjacent to the windbreak), and the levels of chlorophylls, malondialdehyde, major phytohormones as well as the activities of peroxidase (POD) and superoxide dismutase (SOD) were also measured. The results showed the content of total chlorophylls and the activity of POD in mature tea leaves gradually decreased with the distance to windbreak, while SOD showed the opposite. The major phytohormones were highly accumulated in the moderately cold-injured tea leaves. The biosynthesis of abscisic acid (ABA) was enhanced in the moderate cold damaged tea leaves, suggesting that ABA plays an important role in the cold response and resistance of tea plants. The transcriptomic result showed that the samples in different rows were well discriminated, and the pathways of plant-pathogen interaction and flavonoid biosynthesis were enriched based on KEGG analysis. WRKY, GRAS and NAC were the top classes of transcription factors differentially expressed in the different cold-injured tea leaves. Thus, windbreak is effective to protect adjacent tea plants from cold wave, and phytohormones importantly participate in the cold injury recovery of tea leaves.

## 1. Introduction

Tea plant (*Camellia sinensis* (L.) O. Kuntze), a widely cultivated economical crop in China, faces various environmental stresses in the field, such as cold, drought and heat stresses. In the southeast of China, especially around Yangtze River, the young shoots of tea plants are susceptible to low temperature and often suffer from cold injury in early spring. Cold injury greatly reduces the production yield of tea leaves [1], not only bringing a series of immediate damage to tea plants, but also resulting in many sequelae. The previous study reports that cold stress is adverse to the growth of plant through inhibiting enzyme activities, accumulating reactive oxygen species (ROS), disturbing the biosyntheses of chlorophyll and phytohormones [2]. In addition, a number of cold stress-related genes have been characterized in tea plants, for example *CsICE* and *CsCBF*, *CsHis*, *CsbZIP*, *CsWRKY2*, *CsARF* and *CsBAM3* [3,4,5,6,7,8]. 

The ability of plant to recover from cold injury is critical for surviving a cold spell, many physiological and biochemical processes are affected by cold injury and even last for a long time after cold spell, which lead to the adverse or delaying effect on production yield or growing period of crops [9,10]. To date, most studies focused on the physiological changes and stress responses of plants shortly after cold injury [10,11], while the physiological activities and molecular mechanisms of post-cold response are still unclear, despite its important role in the regulation of growth. Phytohormones, such as abscisic acid (ABA), cytokinin (CK), gibberellins (GAs), auxin (IAA), and jasmonic acid (JA), have been reported to be involved in the modulation of cold resistance and recovery of plants [11,12,13]. The information about the molecular mechanisms of cold-injured tea plants in recovery are scarce. 

A windbreak (shelterbelt) is commonly used in agricultural production, providing shelter from the wind, erosion control, and wildlife habitat [14]. A windbreak is also applied to protect tea plants from cold injury, and the protective effect is related to the distance from tea plants to windbreak. In practice, the tea plants closer to windbreak are better protected from cold injury than those far away from windbreak, which means there is an effective range of windbreak. This is verified by the increasing damaged area of tea leaves along with the distance from a windbreak. Thus, the tea plants adjacent to a windbreak have gradually enhanced cold damage of tea leaves due to the reduced protective effect of windbreak along with distance, which are good subjects for investigating the recovery mechanisms of cold injury in tea leaves owing to the increasing cold injury degree. In the present work, the cold injury status of tea plants with different distance from a windbreak were surveyed, and the cold-injured mature tea leaves were submitted to transcriptomic analysis. Besides, the stress-related physiological indicators, including chlorophyll content, malondialdehyde (MDA) level, peroxidase (POD) and superoxide dismutase (SOD) activities, as well as phytohormone levels were measured. The results of this study would provide a novel insight into the recovery mechanisms of tea plants from cold injury.

## 2. Results

### 2.1. The Effects of Windbreaks on the Tea Leaves

Figure 1 shows the tea rows adjacent to the windbreak, including a general view and detailed images. The cold damage degree of tea plants increased with the distance away from the windbreak, with generally increasing injured brown area of mature leaves (Figure 1). The tea plants in Rows 2 and 5 were barely affected, while the tea plants in Rows 8 and 11 were injured. Besides, the delayed growth status of spring buds was observed in Rows 8 and 11 (Figure 1B), suggesting that the growth of new shoots was retarded due to cold damage. 

Figure 2 shows the stress-related indicators of tea leaves collected from different rows, including chlorophyll content, MDA level, as well as the activities of POD and SOD. The content of chlorophylls in the mature tea leaves gradually decreased with the distance away from the windbreaks, which is consistent with the field observation in Figure 1. The samples in Row 2 contained the highest content of chlorophyll (1.21 ± 0.04 mg/g), subsequently followed by Row 5 (1.05 ± 0.03 mg/g) and Row 8 (0.87 ± 0.11 mg/g), while the samples in Row 11 contained the lowest content of chlorophyll at 0.71 ± 0.02 mg/g. The highest level of MDA (17.3 ± 0.61 μg/g) was obtained in Row 2, which was significantly higher than those in other rows (14.0 ± 1.23~15.4 ± 1.04 μg/g, Figure 2B). The activities of POD in Rows 2 and 5 were 222 U/g and 205 U/g, being significantly higher than 159 U/g and 110 U/g of Row 8 and Row11, while the activities of SOD showed an opposite change trend (Figure 2C,D). Relatively higher activities of SOD were obtained in the samples of Rows 8 and 11, being 27,379 U/g and 28,924 U/g respectively, compared with 21,667 U/g and 21,048 U/g of Rows 2 and 5. 

### 2.2. The Levels of Phytohormones in the Cold-Injured Tea Leaves during Recovery Period

Figure 3 shows the contents of the major phytohormones in tea samples. The contents of abscisic acid (ABA), gibberellins (GAs), *trans*-Zeatin-riboside (TZR), and *trans*-zeatin (TZ) in the mature tea leaves gradually increased with the distance away from the windbreak (from Row 2 to Row 8), and the highest levels of these phytohormones were observed in the samples of Row 8. By contrast, the contents of jasmonic acid (JA) decreased with the distance away from the windbreak, while the contents of IAA and n6-(2-isopentenyl)-adenine (2-IP) showed fluctuant trends, with the lowest contents being observed in Row 11. The lowest contents of many phytohormones were observed in the samples of Row 11, including ABA, GA1, GA3, TZ, 2-IP, JA and IAA. A general decline of JA was observed in the tea samples as the distance to the windbreak increased (Figure 3), while the highest content of IAA was observed in the samples of Row 2 (420.30 pg/g), compared with its lowest content in Row 11 (238.27 pg/g). IAA level is closely related with the growth of plants. 

### 2.3. The Transcriptomic Profiles of Tea Leaves in Different Rows

The statistics on the RNA-seq data is shown in Appendix A. The RNA-seq libraries of different tea samples were prepared and sequenced on the HiSeq platform using 150 bp paired-end sequencing. As a result, 41.3~46.0 million, 38.1~52.2 million, 40.3~49.4 million, and 39.4~50.6 million RNA-seq clean reads were respectively obtained for Rows 2, 5, 8 and 11, with 71.79~86.41% of genes matched to reference genome (Appendix A). The gene expression data was presented in Appendix A, which was further validated by reverse transcriptase-quantitative polymerase chain reaction (RT-qPCR) analysis. Appendix A showed the qPCR results were highly correlated with the corresponding transcriptomic data (R^2^ = 0.8174), suggesting that the transcriptomic dataset was able to represent the transcript abundances. 

The PCA score plot of RNA-seq data showed that the tea samples collected from different rows were clustered and discriminated from each other (Figure 4A). The first two principal components (PC) explained 94.0% of the total variation (PC1 = 77.1%, PC2 = 16.9%). Figure 4B displayed the numbers of up- and down-regulated different expressed genes (DEGs) in comparison pairs. The pair of Row 5 vs Row 2 had the lowest number of DEGs (153 up-regulated and 511 down-regulated), while the pair of Row 11 vs. Row 8 had the highest number of DEGs (1829 up-regulated and 2375 down-regulated). This result was in an agreement with the distributions of tea samples in the PCA score plot (Figure 4A). 

Figure 4C showed the enriched top 10 KEGG pathways of DEGs for different comparison pairs. More pathways were significantly enriched in the pairs of Row 11 vs. Row 8 and Row 8 vs. Row 5, compared with the pair of Row 5 vs. Row 2, suggesting that the sample difference among Row 11, Row 8 and Row 5 was greater than that between Row 5 and Row 2. The pathways of plant-pathogen interaction and flavonoid biosynthesis were enriched in these three comparison pairs, while the pathways of photosynthesis-antenna proteins and photosynthesis were enriched in the comparison pair of Row 11 vs. Row 8. 

### 2.4. The Profiles of Transcription Factors (TFs) in Differentially Cold-Injured Tea Leaves

The TFs were annotated according to the transcription factors database in Tea Plant Information Archive (http://tpdb.shengxin.ren/, (accessed on 15 March 2022). To examine the profiles of TFs in differentially cold-injured tea leaves, we selected the TFs with FPMK value being above 20 in all the tea samples. There were 262 TFs obtained (Appendix A), including WRKY, GRAS and NAC transcription factor classes. Figure 5A shows WRKY were the most abundant TFs, accounting for 6.1% of the total selected TFs, subsequently followed by GRAS (5.3%), C3H (5.3%), NAC (5.0%), C2H2 (4.6%), bZIP (4.6%), bHLH (4.2%), AP2/ERF (4.2%) and MYB (3.4%). Most of these TFs, including WRKY, GRAS, MYB, C3H, C2H2, NAC, bZIP, bHLH, AP2/ERF involved in the plant resistance to cold stress, showed dynamic changes in the tea plants from different rows (Figure 5B) 

### 2.5. The Biosynthesis Pathways of ABA, JA and IAA

Figure 6 shows the biosynthesis pathways of ABA, JA and IAA in differentially cold-injured tea leaves. For ABA, β-cryptoxanthin are produced from β-carotene through the enzymatic reaction of β-carotene 3-hydroxylase (crtZ) and β-carotene hydroxylase (LUT5), which were further channeled into the biosynthesis of ABA through zeaxanthin epoxidase (ZEP), 9-*cis*-epoxycarotenoid dioxygenase (NCED), xanthoxin dehydrogenase (ABA2), and abscisic-aldehyde oxidase (AAO3). Relatively higher expression levels of ABA biosynthetic structural genes were achieved in Row 8 and Row 5 (Figure 6), which was in a general agreement with the content of ABA in different tea samples (Figure 3), while the biosynthesis of ABA was attenuated in Row 2 and Row 5. The biosynthesis of JA is comprehensively regulated by several structural genes, including lipoxygenase (LOX2S), hydroperoxide dehydratase (AOS), allene oxide cyclase (AOC), 12-oxophytodienoic acid reductase (OPR), OPC-8:0 CoA ligase 1 (OPCL1), acyl-CoA oxidase (ACX), enoyl-CoA hydratase/3-hydroxyacyl-CoA dehydrogenase (MFP2), and acetyl-CoA acyltransferase ACAA1 (Figure 6). Our result showed that the transcriptions of upstream genes, such as LOX2S, AOS, AOC, OPR, OPCL1, were up-regulated in the well-protected Rows 2 and 5, while only ACX and ACAA1 were down-regulated in Row 2 and Row 5. This was generally in line with the JA level in different tea samples that Row 2 contained the highest level of JA. IAA is synthesized from L-tryptophan via the catalysis of tryptophan-pyruvate aminotransferase (TAA1) and indole-3-pyruvate monooxygenase (YUCCA). The highest expressions of TAA1 and YUCCA were obtained in Row 2, which was consistent with the highest level of IAA in Row 2 (Figure 3). 

## 3. Discussion

Chlorophyll content is regarded as an indicator of the functional changes of photosynthetic apparatus under temperature stress [15]. Especially for tea plants, tea leaves turn brown as cold damage occurs. This is because ice crystals form within the tissues of tea leaves due to coldness, which damage the structure of cell membrane and trigger the enzymatic reactions of polyphenols resulted in the brown color of tea leaves. Hence, the chlorophyll content of tea leaves is negatively correlated with the brown area of cold injured leaves, which is an effective indicator for the cold damage degree of tea plants. Malondialdehyde (MDA) is a product of lipid peroxidation. The level of MDA in plants is an indicator of stress that evaluates the damage degree [16]. Peroxidase (POD) widely present in plants is regarded as a defense antioxidant, scavenging hydrogen peroxide (H_2_O_2_) through oxidation of co-substrates like phenolic compounds [17]. The change of POD activity is generally related to cold stress response [18]. Superoxide dismutase (SOD) also plays an important role in the enzymatic antioxidant system in plants, scavenging free radicals and reactive oxygen species (ROS) generated in response to stress [19,20]. Stress responses in tea plants involve antioxidant defense system to alleviate the harmful effects of ROS. The redox homeostasis in plants is maintained by two classes of oxidation resistance, antioxidant enzymes and non-enzymatic low molecular antioxidants, such as flavonoids, ascorbic acid, and reduced glutathione [21]. SOD is an important antioxidant enzyme in plants [20]. In this study, the SOD activities of Row 8 and Row 11 were significantly higher than those of Row 2 and Row 5 (Figure 2), indicating that the tea plants in Row 8 and Row 11 might still suffer from the cold injury-induced oxidative stress. This suggests that the windbreaks indeed provided shelter on the adjacent tea plants, although the protective effect gradually decreased with the distance to windbreak. By contrast, the activity of POD gradually decreased from Row 2 to Row 11, suggesting that POD and SOD might play different roles in the antioxidant defense system of tea plants. The opposite change trend of SOD and POD was also observed in the chilling-stressed anthurium [22]. The protection effect of windbreak on paddock microclimate was reported, in which the shelterbelt reduced wind speed and decreased the loss of temperature and humidity [23]. 

Phytohormones are crucial to the growth, development, and stress resistance of plants [24,25,26]. Our results indicated that the biosynthesis of the major phytohormones in mature tea leaves were affected after cold injury. The highest levels of most phytohormones were obtained in Row 8, except JA and IAA. This suggests that biosynthesis of JA and IAA may be varied in response to different degree of cold stress. ABA and GAs are important phytohormones associated with the adaptation of plants to cold stress [27,28]. The highest levels of ABA and GAs as well as the promoted biosynthesis of ABA were observed in Row 8, which is consistent with the observation that cold-injury induced the genes expression over ABA and GAs biosynthesis [11,12]; while the Row 11 contained the lowest level of ABA, which was probably attributed to the cold-injury induced biosynthetic disfunction of ABA in severely damaged tea leaves. IAA promotes the growth of plants [29]. Row 2 contained the highest level of IAA, which was in line with its early development and strongest body of young shoots in Figure 1. 

TFs are crucial regulators for plants to respond to or accommodate to abiotic and biotic stress. WRKY proteins have been reported to participate in the cold stress response of plants, such as eggplant [30]. In the present study, WRKY accounted for the highest proportion of the total selected TFs differentially expressed in different cold-injured tea leaves. Many GRAS (gibberellic acid insensitive, repressor of GAI, and scarecrow) genes can respond to cold stress and function in leaves [31]. From Row 2 to Row 11, the expressions of GRAS genes in our study generally decreased, suggesting that GRAS might participate in the recovery of tea plant after cold stress. C2H2 and C3H TFs are involved in the cold stress tolerance mechanisms in plants [32]. Figure 5B showed most of C2H2 and C3H TFs had the lowest transcription levels in Row 11 with the highest cold damage degree, suggesting that the TFs belonging to C2H2 and C3H families are likely to be involved in the resistance of tea plants. In tea plants, several bZIP TFs (e.g., CsbZIP29, CsbZIP52, and CsbZIP08) have been reported to be involved in the response to cold stresses [33]. Our result also showed the expressions of bZIP TFs varied with the cold damage degree of mature tea leaves. The expression levels of bHLH TFs in Row 2 and Row 5 were roughly higher than those of Row 8 and Row 11 (Figure 5B). This is attributed to the better protection of the windbreaks on the adjacent tea rows, resulting in the better resistance of Row 2 and Row 5 that were hardly harmed by the cold wave. The correlation between over-expressed bHLH TFs and cold stress tolerance has been reported in *Arabidopsis* [34]. AP2/ERF is also a large TF family in plants associated with cold stress. The important members of AP2/ERF involved in cold stress are CBFs, acting as pioneers of plant regulatory hub genes in response to cold stress. The overexpression of CBF homologs in many plants up-regulated the expression of cold-relevant genes [35]. The expressions of AP2/ERF TFs gradually decreased from Row 2 to Row 11, suggesting that many AP2/ERF TFs may be related with the resistance of tea plants. Moreover, MYB TFs play multiple roles in mediating the expression of CBFs in cold-stressed plants, such as *Arabidopsis* [36]. The lowest transcriptions of MYB TFs were obtained in Row 11 that had the highest cold damage degree (Figure 5B). Our result indicated the change in the expressions of MYB TFs generally coordinated the expressions of AP2/ERF TFs.

## 4. Materials and Methods

### 4.1. Tea Plant Materials and Sampling

The tea plants ‘Zhenong 139’ were grown in the Chishan Lake organic tea plantation, Lanxi, Zhejiang Province, China (N 29°11′, E 119°48′). From 29 December 2020 to 1 January 2021, the tea plants ‘Zhenong 139′ suffered from cold wave (the lowest temperature down to −5 °C, the wind power up to level 3), which had severe cold damages, especially tea leaves. Appendix A is the local weather report from 1 November 2020 to 3 March 2021. On 3 March 2021, the second mature leaves basipetal from the apical tip were collected from the second, fifth, eighth and eleventh rows (termed as Row 2, Row 5, Row 8 and Row 11), which were approximately 3.0 m, 7.5 m, 12 m, and 16.5 m away from the windbreaks. Three independent biological replicates were applied for each sample. The collected tea samples were immediately placed in liquid nitrogen and stored at −80 °C for further analyses. 

### 4.2. Transcriptomic and Bioinformatic Analyses

The isolation and sequencing of mRNA were conducted by Gene Denovo Biotechnology Co., Ltd. (Guangzhou, China). In brief, the total RNA was extracted using Trizol reagent kit (Invitrogen, Carlsbad, CA, USA), and then was enriched by Oligo (dT) beads and fragmented. The reverse-transcription of obtained fragments and then second-strand cDNA synthesis were performed using NEBNext Ultra RNA Library Prep Kit (New England Biolabs, Ipswich, MA, USA). After purification by QiaQuick PCR extraction kit, end repair, and polyA addition, the cDNA was ligated to Illumina sequencing adapters, then the ligated products were selected by agarose gel electrophoresis, PCR amplified, and purified with AMPure XP beads to obtain the library. The sequencing was performed on Illumina HiSeqTM 2500. The raw reads were filtered by fastp (version 0.18.0) to obtain the high-quality clean reads through removing adaptor, duplication, ambiguous sequences (reads with higher than 10% “N” rate), and low quality reads containing more than 50% of low quality (Q-value ≤ 20) bases. The obtained clean reads were blasted against the reference tea genome of ‘Shuchazao’ using HISAT2.2.4 with default set. Fragments Per Kilobase of transcript per Million mapped reads (FPKM) method was used to normalize a gene expression level using StringTie v1.3.1 software. The differentially expressed genes (DEGs) were identified by DESeq2 under the criteria of (ǀlog_2_ fold changeǀ > 1, FDR < 0.05) based on read counts. The DEGs were blasted against the online Kyoto Encyclopedia of Genes and Genomes (KEGG) database (http://www.genome.jp/tools/blast/, (accessed on 8 July 2021) to extract the enriched metabolic pathways. 

### 4.3. Determination of Chlorophyll and Malondialdehyde (MDA)

The chlorophyll assay kit (Nanjing Jiancheng Bioengineering Institute, Nanjing, China) was used to determine the content of total chlorophylls in different tea samples. Briefly, the tea leaves (0.1 g) were ground in liquid nitrogen with 1 mL distilled water and 50 mg absorbent in dark, 10 mL of extraction solution [ethanol:acetone = 1:2 (*v*/*v*)] was added and kept until the sediment turned white in dark. The extract was filtered and the filtrate was analyzed using Epoch microplate spectrophotometer (Bio Tek, Winooski, VT, USA) at the wavelengths of 645 and 663 nm. 

The plant malondialdehyde (MDA) assay kit (colorimetric method) was purchased from Nanjing Jiancheng Bioengineering Institute, Nanjing, China. The tea leaves (0.1 g) were ground in liquid nitrogen, then the powder was transferred to a 1.5 mL centrifuge tube, followed by the addition of 0.9 mL extraction buffer. After homogenization, the supernatant was centrifuged at 8000 rpm and 4 °C for 10 min. Further, the supernatant reacted with thiobarbituric acid (TBA) according to the protocol, and the absorbance of mixture was analyzed by Epoch microplate spectrophotometer (Bio Tek, Winooski, VT, USA) at the wavelength of 532 nm.

### 4.4. Determination of SOD and POD Activities

The activities of SOD and POD were measured by superoxide dismutase (SOD) assay kit (WST-1 method) and peroxidase (POD) assay kit (Nanjing Jiancheng Bioengineering Institute, Nanjing, China). Briefly, the tea leaves (0.2 g) were ground in liquid nitrogen, followed by the addition of 1.8 mL phosphate buffer (pH = 7.4). The homogenate was centrifuged at 15,000 rpm and 4 °C for 20 min. The supernatant was collected for reaction according to the protocols, and the absorbance of reactant was analyzed by Epoch microplate spectrophotometer (Bio Tek, Winooski, VT, USA). For SOD activity, the absorbance of the mixture was measured at the wavelength of 450 nm. For POD activity, the absorbance of the mixture was measured at the wavelength of 420 nm.

### 4.5. Analysis of Phytohormones

The analysis of phytohormones was conducted by Gene Denovo Biotechnology Co., Ltd. (Guangzhou, China). The method was described in our previous work [37]. Briefly, the tea samples were taken out from −80 °C storage, then weighted, ground and placed in 10 mL centrifuge tube (with plug), and 5 mL of extraction solution [methanol:water:formic acid = 15:4:1 (*v*/*v*)] containing 0.5% of butylated hydroxytoluene was added. The mixture was ultrasonicated for 30 min, followed by an incubation at −40 °C for 60 min. After centrifugation at 12,000 rpm and 4 °C for 10 min, the supernatant was collected and submitted to solid phase extraction at the flow rate of 1 mL/min. After rinsing, the Sep-Pak C18 3 cc Vac Cartridge (Waters Corporation, Milford, MA, USA) column was eluted using 1 mL methanol. The eluate was evaporated in vacuum and reconstituted with 400 μL of 80% methanol solution. After centrifugation at 12,000 rpm and 4 °C for 10 min, the supernatant was submitted to LC-MS/MS (Waters Acquity UPLC and AB SCIEX 5500 QQQ -MS) analysis. The UHPLC conditions were as follows: Acquity UPLC HSS T3 (1.8 µm, 2.1 mm × 100 mm), column temperature 30 °C, injection volume 15 μL, phase A = 0.1% formic acid solution, phase B = acetonitrile, the gradient elution program starting from 90% A/10% B for the first 1 min, to 30% A/70% B at 3 min, to 10% A/90% B at 5 min and maintaining at 10% A/90% B for another 3 min, followed by re-equilibrium for 2 min, flow rate 0.30 mL/min. MS condition was: ESI ion source, turbo spray; source temperature 450 °C; ion spray voltage (IS) 5500 V (positive ion mode)/−4500 V (negative ion mode); the collision gas (CAD) was high. A multiple reaction monitoring (MRM) mode was used for quantifying ABA, GAs, TZR, 2-IP, TZ and JA. Data acquisition, peak integration, and calculations were carried out by MultiQuant software. All of the phytohormones were quantified by calculating the area of individual peak, using authentic phytohormone standards. The work solutions of different phytohormones (200 ng/mL) were prepared using 80% methanol aqueous solution. Then, the standard solutions were prepared through dilution to a series of concentrations (0.05, 0.1, 0.2, 0.5, 1.0, 2.0, 5.0, 10.0, 20.0 ng/mL), using 80% methanol aqueous solution. The standard curve for each phytohormone was obtained, with the correlation coefficient R^2^ being above 0.99.

### 4.6. RT-qPCR Validation

RT-qPCR analysis were performed according to the previous method [38]. Briefly, the first-strand cDNA was synthesized from 1 μg of total RNA, using HiScript^®^ II Q RT SuperMix for qPCR (+gDNA wiper) (Vazyme Biotech Co., Ltd., Nanjing, China). Appendix A shows the list of specific primers designed on Primer 3 software based on the genome sequences of tea plant genome sequences of *Camellia sinensis* cv. Shuchazao, using β-actin as an internal control. The qPCR cycling was performed on Applied Biosystems™ StepOnePlus™ Real-Time PCR System (Applied Biosystems™ ABI, Carlsbad, CA, USA) based on the introduction of ChamQ SYBR qPCR Master Mix (High ROX Premixed) (Vazyme Biotech Co., Ltd., Nanjing, China): 95 °C for 30 s and 40 cycles at 95 °C for 10 s, annealing at 60 °C for 30 s. The relative expression level of each gene was calculated by 2 ^−∆∆Ct^ method. Technical triplicates were employed for each biological replicate.

### 4.7. Data Analysis

All the tests were repeated three times and presented by mean value ± SD. The significant difference analysis was carried out by the SAS System for Windows version 8.1 (SAS Institute Inc., Cary, NC, USA), using a Tukey test. Principal component analysis (PCA) based on a correlation matrix was conducted by Minitab 17 statistical software (Minitab. LLC, State College, PA, USA). The heatmap was plotted using z-score values of transcriptomic dataset and drafted on an online platform of OmicShare tools (http://www.omicshare.com/tools, (accessed on 23 March 2022).

## 5. Conclusions

The protective effect of windbreak on tea plants was verified in the present study, in terms of biochemical indicators and the transcriptional levels of relevant genes. Cold wave caused damage to mature tea leaves, which caused the lower content of total chlorophylls and altered activities of POD and SOD in the cold injured tea leaves. The highest contents of major phytohormones were obtained in Row 8, suggesting that more complicated physiological activities occurred in the moderate cold-injured tea leaves, compared with the well-protected and the badly damaged tea leaves. The biosynthesis of ABA was enhanced in the moderate cold damaged tea leaves, while the biosynthesis of IAA was attenuated, suggesting that ABA plays an important role in the cold response and resistance of tea plants. WRKY, GRAS and NAC are important classes of transcription factors involved in the cold injury of tea leaves. This study gives new insights into the cold injury mechanisms of tea plants after cold wave, and also gives scientific guide for developing appropriate agronomic measures to protect tea plants from cold damage.

## Figures and Tables

**Figure 1 plants-11-02751-f001:**
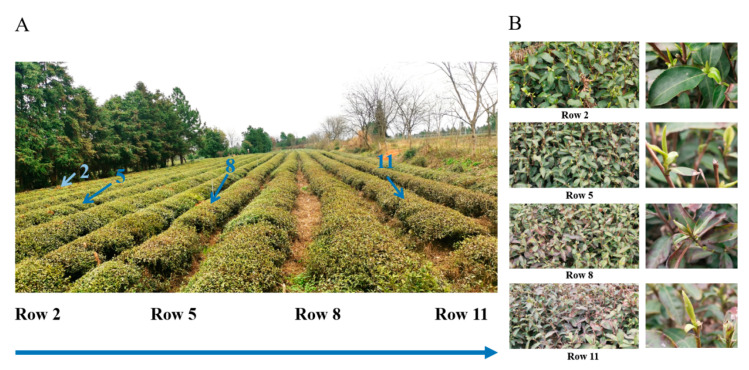
The tea rows adjacent to the windbreak: (**A**) General view and (**B**) Detailed images of tea plants in each row.

**Figure 2 plants-11-02751-f002:**
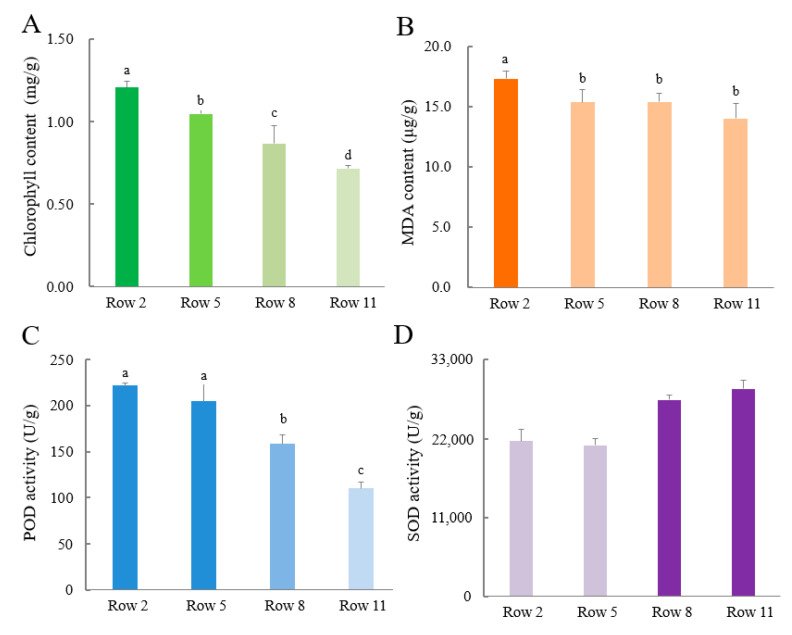
The stress-related indicators of tea leaves collected from different rows: (**A**) Chlorophyll content, (**B**) MDA level, (**C**) POD activity and (**D**) SOD activity. Data with different letters (a, b, c, d) were significantly different at *p* < 0.05.

**Figure 3 plants-11-02751-f003:**
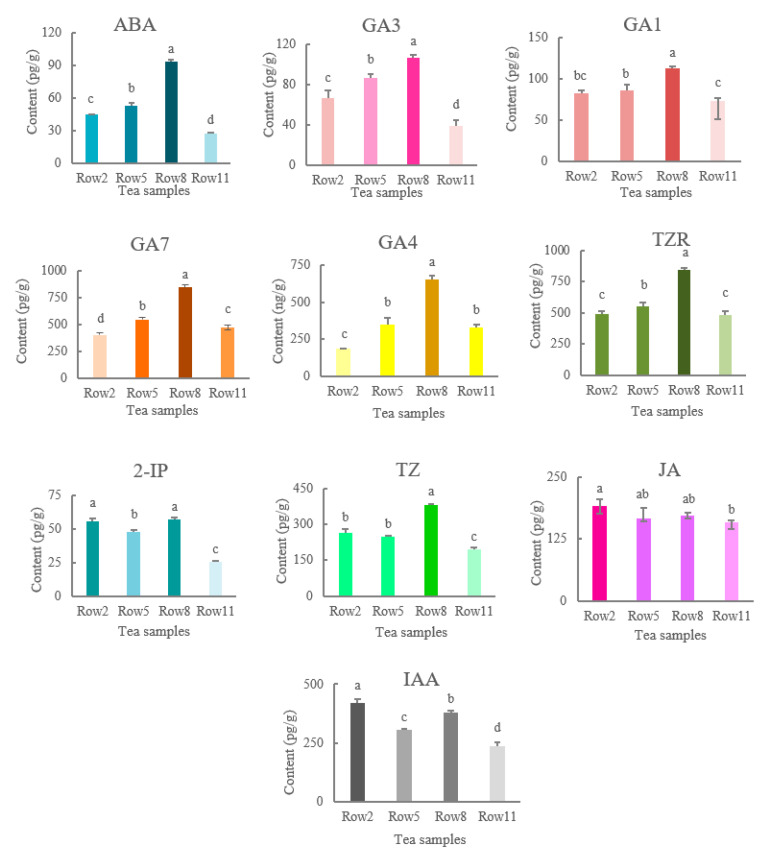
The contents of the major phytohormones in the mature tea plants grown in different rows (from 2 to 11 in terms of distance to windbreak). ABA: abscisic acid; GA: gibberellins; TZR: *trans*-Zeatin-riboside; TZ: *trans*-zeatin; 2-IP: n6-(2-isopentenyl)-adenine; JA: jasmonic acid; IAA: auxin. Data with different alphabetic letters (a, b, c, d) were significantly different at *p* < 0.05.

**Figure 4 plants-11-02751-f004:**
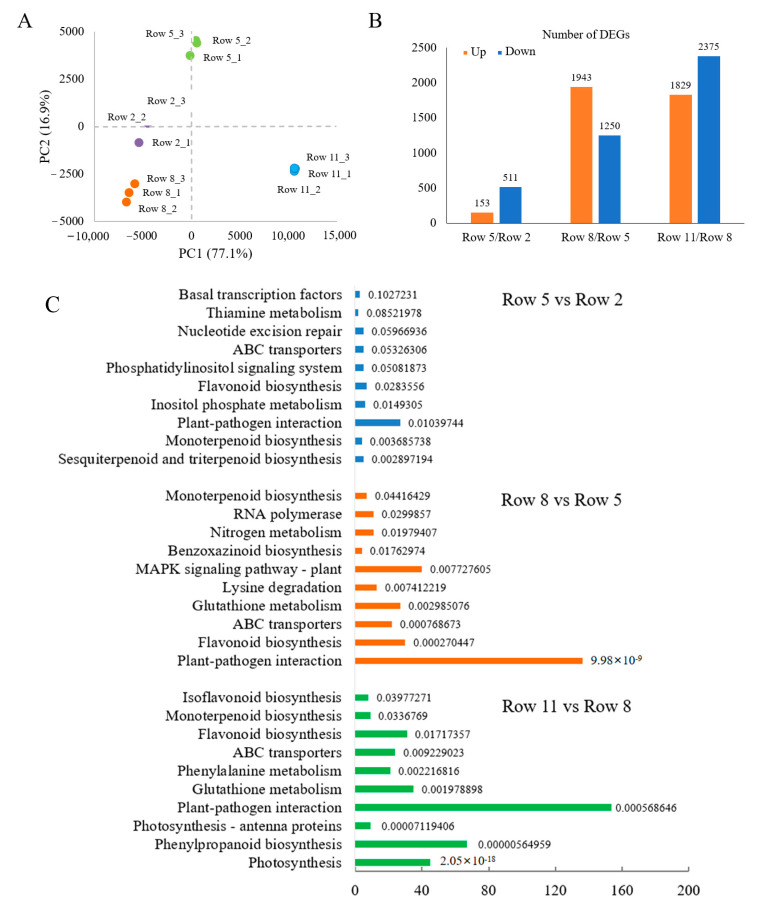
The gene expression profiles of different tea samples. (**A**) PCA analysis; (**B**) DEG numbers; (**C**) Top 10 KEGG pathways of DEGs with the annotation of *p*-value.

**Figure 5 plants-11-02751-f005:**
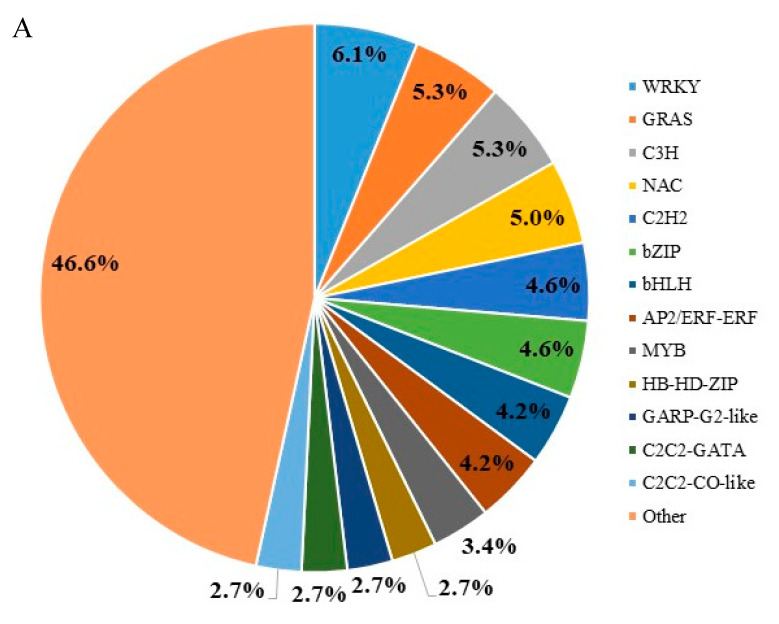
The proportions of different classes of TFs with FPKM value being above 20 (**A**) and the heatmap of transcript abundances of TFs (**B**).

**Figure 6 plants-11-02751-f006:**
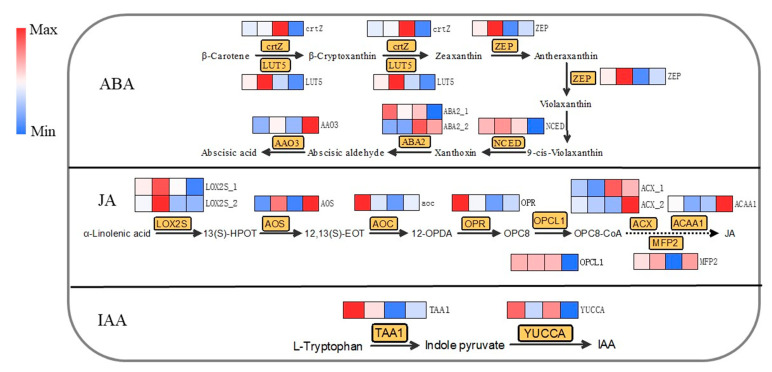
The biosynthesis of ABA, JA and IAA in differentially cold-injured mature tea leaves. The heatmap was plotted using log_2_ transformed values of RNA-seq dataset, and the cells in the heatmap from left to right represented Row 2, Row 5, Row 8 and Row 11. crtZ: β-carotene 3-hydroxylase; LUT5: β-carotene hydroxylase; ZEP: zeaxanthin epoxidase; NCED: 9-*cis*-epoxycarotenoid dioxygenase; ABA2: xanthoxin dehydrogenase; AAO3: abscisic-aldehyde oxidase; LOX2S: lipoxygenase; AOS: hydroperoxide dehydratase; AOC: allene oxide cyclase; OPR: 12-oxophytodienoic acid reductase; OPCL1: OPC-8:0 CoA ligase 1; ACX: acyl-CoA oxidase; MFP2: enoyl-CoA hydratase/3-hydroxyacyl-CoA dehydrogenase; ACAA1: acetyl-CoA acyltransferase; TAA1: tryptophan-pyruvate aminotransferase; YUCCA: indole-3-pyruvate monooxygenase.

## Data Availability

The RNA-seq raw data (Accession number: CRA006758) were up-loaded to BIG data center (https://bigd.big.ac.cn/, accessed on 25 April 2022) under the project of No.PRJCA009174.

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
