# Peer review of "Integrative Transcriptomic and Phytohormonal Analyses Provide Insights into the Cold Injury Recovery Mechanisms of Tea Leaves"

_plants, 2022, doi:10.3390/plants11202751_

Round 1

Reviewer 1 Report

Dear authors,

please find attached the revised MS.

Overall, methods are well-presented and consistent with the experimental design. Please note comment in results regarding the presence of "discussion" material in the results chapter.

Author Response

Dear authors,

please find attached the revised MS.

Overall, methods are well-presented and consistent with the experimental design. Please note comment in results regarding the presence of "discussion" material in the results chapter.

Response: The manuscript was revised accordingly.

According to the format of the MS, this should be Results and Discussion. Plants allows for this style in MS layout. Please consider modify the overall layout, otherwise a proper Discussion chapter is needed.

Response: According to the word template of Plants, the sections of “Results” and “Discussion” should be separated. We have moved the discussion sentences from Result Section to Discussion Section.

Reviewer 2 Report

In the manuscript ‘Integrative transcriptomic and phytohormonal analyses provide insights into the cold injury recovery mechanisms of tea leaves’ Ni and collaborators tried to analyze the physiological activities and molecular mechanisms of post-cold response in tea plants.

The abstract should be revised. I would suggest removing ‘Row 8’ from the abstract, and similar terms throughout the text, and replaced them by more appropriate terms.

The Introduction section could have more information about the research topic and on previous studies. The experimental design is correct and the methodologies are well described. The Results are clearly presented. I think a more in-depth discussion would be helpful. The same relationship between the gene expression and it association with cold injury resistance is made by the authors several times (ex: Lines 226, 228, 239). The authors should explain why a decrease in the abundance of a certain transcript imply an immediate association of the gene with the recovery to cold injury? How do authors evaluate that those plants, in row 11, were in the recovery state? In summary, there are many assertions made by the authors which are not deeply substantiated. Future work was not presented.

Author Response

In the manuscript ‘Integrative transcriptomic and phytohormonal analyses provide insights into the cold injury recovery mechanisms of tea leaves’ Ni and collaborators tried to analyze the physiological activities and molecular mechanisms of post-cold response in tea plants.

The abstract should be revised. I would suggest removing ‘Row 8’ from the abstract, and similar terms throughout the text, and replaced them by more appropriate terms.

Response: The abstract was revised, and we prefer to use the number of rows to indicate tea samples for simplification, because the distance to the windbreak for each row has been added to the Method section. ‘Row 8’ has been removed from the abstract.

The Introduction section could have more information about the research topic and on previous studies. The experimental design is correct and the methodologies are well described. The Results are clearly presented. I think a more in-depth discussion would be helpful. The same relationship between the gene expression and it association with cold injury resistance is made by the authors several times (ex: Lines 226, 228, 239). The authors should explain why a decrease in the abundance of a certain transcript imply an immediate association of the gene with the recovery to cold injury? How do authors evaluate that those plants, in row 11, were in the recovery state? In summary, there are many assertions made by the authors which are not deeply substantiated. Future work was not presented.

Response: The manuscript was revised accordingly. This is a comparative study. Since the distance to the windbreak is the most important environmental factor affecting the growth of these tea plants, resulting in the gradually changing cold injury degree of tea leaves due to cold wave. Thus, the abundant changes of certain transcripts could be associated with the cold injury recovery of tea plants in response to different cold injury degrees. Since the cold wave occurred from December 29th, 2020 to January 1st, 2021, the sampling time was March 3rd, 2021. Based on the weather report from February 22nd, 2021 to March 3rd, 2021 in Table S4 (No obvious cold threat), we consider the tea leaves should be in the recovery state rather than under cold stress. The discussion has been revised, and future work has been presented.

Reviewer 3 Report

The authors present an interesting study about the differential responses to cold stress in tea plants growing at different distances from windbreaks. The authors analyzed the content of chlorophyll, antioxidant enzymes, hormones and the expression of TFs and other genes related to stress. The study bring good results, however, several corrections, clarifications and detailments must be done before the acceptance by Plants.

My main concern is about the ensurance that the responses measured are due to the cold and not to other stress. The cold wave happened in January, while the harvesting of the leaves was conducted in March (2 months latter). How can the authors ensure that no other stresses happened during this time? Besides, the M&M, results and discussion sections must be carefully checked. Some details are missing, some parts are not clear, the numbers of some figures are not right and several discussion parts are merged in the results section. In addition, the discussion can be improved. All my detailed comments and suggestions can be found in attached PDF. 

Author Response

The authors present an interesting study about the differential responses to cold stress in tea plants growing at different distances from windbreaks. The authors analyzed the content of chlorophyll, antioxidant enzymes, hormones and the expression of TFs and other genes related to stress. The study bring good results, however, several corrections, clarifications and detailments must be done before the acceptance by Plants.

My main concern is about the ensurance that the responses measured are due to the cold and not to other stress. The cold wave happened in January, while the harvesting of the leaves was conducted in March (2 months latter). How can the authors ensure that no other stresses happened during this time? Besides, the M&M, results and discussion sections must be carefully checked. Some details are missing, some parts are not clear, the numbers of some figures are not right and several discussion parts are merged in the results section. In addition, the discussion can be improved. All my detailed comments and suggestions can be found in attached PDF. 

Response: Cold stress is the primary stress to tea plants during the winter, although other stress can not be ruled out. However, this is a comparative study. The tea plants with increasing distance to the windbreak had increasing cold injury degree due to the decreasing protective effect of windbreak, while might suffer from other stress by the same degree. The different cold injury degrees of different tea plants were verified by the content of chlorophyll. Thus, the differential profiles of transcriptome and phytohormone in tea plants were mainly attributed to their different cold injury degrees. The M&M, results and discussion sections have been revised accordingly. The manuscript was revised according to the comments annotated in PDF.

The authors must include some climatic measurements (light, temperature and rainfall), since several conditions can affect the parameters analyzed. If it is a manual irrigation, please provide the conditions used. The authors must give information in ºC about what they call as "cold wave". In addition, they must also present the temperature in each row in the moment of harvesting and in the moment of the cold stress, in order to show that the different rows had different temperatures.

Response: The basic weather report was provided in Table S4. The temperature in each row should be the same at the moment of harvesting, due to the gentle breeze and the short distance between different rows. 

Why did the authors choose this genotype? Please clarify.

Response: Actually, we didn’t select the tea cultivar ‘Zhenong 139’. We found the windbreaks in the Northwest of tea plantation (Chishan Lake organic tea plantation, Lanxi, Zhejiang Province), and ‘Zhenong 139’ is the tea plants adjacent to the windbreaks.

Round 2

Reviewer 2 Report

The authors addressed appropriately my comments and the manuscript was significantly improved.